# Ascorbic Acid Improves Tomato Salt Tolerance by Regulating Ion Homeostasis and Proline Synthesis

**DOI:** 10.3390/plants13121672

**Published:** 2024-06-17

**Authors:** Xianjun Chen, Hongwei Han, Yundan Cong, Xuezhen Li, Wenbo Zhang, Jinxia Cui, Wei Xu, Shengqun Pang, Huiying Liu

**Affiliations:** 1Key Laboratory of Special Fruits and Vegetables Cultivation Physiology and Germplasm Resources Utilization of Xinjiang Production and Contruction Crops, Department of Horticulture, Agricultural College, Shihezi University, Shihezi 832003, China; chenxianjun_0805@163.com (X.C.); xjnykx-h@xaas.ac.cn (H.H.); yundan_cong@163.com (Y.C.); lixuezhen492@163.com (X.L.); zhangwb626@163.com (W.Z.); jinxiacui77@163.com (J.C.); xuwei0412@shzu.edu.cn (W.X.); 2Key Laboratory of Molecular Breeding and Variety Creation of Horticultural Plants for Mountain Features in Guizhou Province, School of Life and Health Science, Kaili University, Kaili 556011, China; 3Key Laboratory of Horticulture Crop Genomics and Genetic Improvement in Xinjiang, Institute of Horticultural Crops, Xinjiang Academy of Agricultural Sciences, Urumqi 830000, China

**Keywords:** ascorbic acid, ion homeostasis, osmotic balance, salt stress, tomato

## Abstract

In this study, processing tomato (*Solanum lycopersicum* L.) ‘Ligeer 87-5’ was hydroponically cultivated under 100 mM NaCl to simulate salt stress. To investigate the impacts on ion homeostasis, osmotic regulation, and redox status in tomato seedlings, different endogenous levels of ascorbic acid (AsA) were established through the foliar application of 0.5 mM AsA (NA treatment), 0.25 mM lycorine (LYC, an inhibitor of AsA synthesis; NL treatment), and a combination of LYC and AsA (NLA treatment). The results demonstrated that exogenous AsA significantly increased the activities and gene expressions of key enzymes (L-galactono-1,4-lactone dehydrogenase (GalLDH) and L-galactose dehydrogenase (GalDH)) involved in AsA synthesis in tomato seedling leaves under NaCl stress and NL treatment, thereby increasing cellular AsA content to maintain its redox status in a reduced state. Additionally, exogenous AsA regulated multiple ion transporters via the SOS pathway and increased the selective absorption of K^+^, Ca^2+^, and Mg^2+^ in the aerial parts, reconstructing ion homeostasis in cells, thereby alleviating ion imbalance caused by salt stress. Exogenous AsA also increased proline dehydrogenase (ProDH) activity and gene expression, while inhibiting the activity and transcription levels of Δ1-pyrroline-5-carboxylate synthetase (P5CS) and ornithine-δ-aminotransferase (OAT), thereby reducing excessive proline content in the leaves and alleviating osmotic stress. LYC exacerbated ion imbalance and osmotic stress caused by salt stress, which could be significantly reversed by AsA application. Therefore, exogenous AsA application increased endogenous AsA levels, reestablished ion homeostasis, maintained osmotic balance, effectively alleviated the inhibitory effect of salt stress on tomato seedling growth, and enhanced their salt tolerance.

## 1. Introduction

Soil salinization poses a significant challenge to agricultural production, impeding crop growth and limiting yield formation [1,2]. Tomato (*Solanum lycopersicum* L.), widely cultivated worldwide, is crucial for global agricultural production and trade. Xinjiang, China’s largest processing tomato production base, is severely threatened by soil salinization. Salt stress reduces the water potential of plant roots, leading to physiological drought and inducing osmotic stress [3]. Excess salts absorbed by roots enter the aboveground parts through transpiration, causing the accumulation of excessive Na^+^ and Cl^−^ in leaves, which compete with ions such as Ca^2+^, K^+^, and Mg^2+^ and disrupt ionic homeostasis [4]. Osmotic stress and ionic imbalance induce the production of reactive oxygen species (ROS), which in turn causes oxidative stress, lipid peroxidation, and irreparable damage to cellular membranes, ultimately impacting plant growth and development [5]. Consequently, reducing Na^+^ accumulation, restoring ionic homeostasis, and maintaining osmotic balance and redox status are crucial for mitigating the adverse effects of salt stress on plants.

During the course of evolution, plants have developed a mechanism to maintain low levels of Na^+^ by actively removing Na^+^ from the cytoplasm. Under salt stress, Na^+^ accumulation occurs, and plant cells sequester Na^+^ into vacuoles via Na^+^/H^+^ antiporters, thereby reducing ion damage [6]. The salt overly sensitive (SOS) signaling pathway regulates ion homeostasis by modulating the activity of Na^+^/H^+^ antiporters in response to salt stress [7]). In addition to inorganic ions, proline (Pro) plays a crucial role as an osmotic regulator in plants under stress conditions. The increase in proline content in plants under salt stress is tightly regulated by enzymes involved in proline metabolism. Proline biosynthesis in plants primarily occurs through the glutamate (Glu) pathway and the ornithine (Orn) pathway, with the former being predominantly catalyzed by Δ1-pyrroline-5-carboxylate synthetase (P5CS) and the latter mainly regulated by ornithine-δ-aminotransferase (OAT) [8]. Most studies indicate that the P5CS pathway is the primary route for proline accumulation during stress [9], while OAT plays a key role in regulating plant cell redox homeostasis by modulating proline metabolism under stress conditions [10].

In recent years, the use of exogenous substances to alleviate salt stress damage in plants has emerged as an effective strategy to enhance plant salt tolerance [11,12,13]. L-ascorbic acid (AsA), acting as an electron donor in redox reactions and an antioxidant, plays crucial roles in plant growth, development, and stress responses [14,15]. AsA, located in the cytoplasm and extracellular space, can directly perceive environmental stressors, thereby regulating antioxidant defense and redox-sensitive signal transduction pathways [16,17]. Moreover, AsA has been shown to effectively enhance plant stress resilience. The application of exogenous AsA can shield lipids and proteins from oxidative damage induced by salt or drought stress [18,19]. Furthermore, numerous studies have underscored the critical role of suitable concentrations of exogenous AsA in ameliorating damage caused by various abiotic stresses, including salt [20], low temperature [21], heavy metals [22,23], and drought [24]. Current research on alleviating salt stress with exogenous AsA primarily focuses on antioxidant defense mechanisms and the mitigation of photoinhibition. For instance, exogenous AsA enhances the activities of antioxidant enzymes in wheat [25], tomato [26], strawberry [20], and cowpea [27] under salt stress, thereby counteracting the adverse effects of salt stress on plant growth. However, research on the regulation of ion homeostasis, selective absorption, and transport, the regulatory mechanism of Na^+^ regulation, and the synthesis and metabolism regulation of proline under salt stress by exogenous AsA remains limited.

Previous studies have demonstrated that exogenous AsA can increase endogenous AsA content [28]. However, it remains unclear whether exogenous AsA increases endogenous AsA levels by regulating the expression and activity of key enzymes involved in AsA biosynthesis. Therefore, in this study, processing tomatoes were used as the experimental material, and exogenous AsA and LYC (lycorine, an inhibitor of AsA synthesis) were applied to create different AsA levels. The study aimed to elucidate the mechanism by which exogenous AsA enhances the salt tolerance of processing tomato seedlings from the perspectives of ion homeostasis and osmotic stress. The results demonstrate that exogenous AsA maintained a high level of endogenous AsA and the redox pool by altering the activities of enzymes related to endogenous AsA synthesis and the expression levels of related genes. It induced the SOS pathway to regulate multiple ion transporters, adjusting intracellular ion homeostasis. Moreover, it enhanced the selective transport of K^+^, Ca^2+^, and Mg^2+^ in the aerial parts, facilitated the efflux and compartmentalization of Na^+^ and Cl^−^, and alleviated ion imbalance caused by salt stress. Additionally, exogenous AsA regulated osmotic adjustment by enhancing Pro degradation and inhibiting its synthesis, thereby mitigating the damage of salt stress to plants. Ultimately, these findings suggest that exogenous AsA effectively enhances root vitality and growth characteristics of processing tomato seedlings under salt stress.

## 2. Materials and Methods

### 2.1. Plant Materials and Treatment Conditions

The experiment was conducted at Shihezi University’s solar greenhouse using the processing tomato cultivar ‘Ligeer 87-5′. Seeds were germinated in plugs with a charcoal and vermiculite mixture (2:1 *v*/*v*). After producing two true leaves, uniform seedlings were transferred to black plastic buckets with foam covers for hydroponics, each containing 10 L of Hoagland nutrient solution (pH = 6.2) diluted with deionized water. After a 7-day pre-cultivation period, seedlings were subjected to various treatments by adding 100 mM NaCl to the nutrient solution for salt stress. Daily leaf sprays of 0.5 mM ascorbic acid (AsA) and 0.25 mM lycorine (LYC), an inhibitor of the key AsA synthesis enzyme L-galactono-1, 4-lactone dehydrogenase (GalLDH), were applied. AsA and LYC were sourced from Sigma (St. Louis, MO, USA) and Yuan Ye (China), respectively. Concentrations were based on prior screening experiments. The five treatments were (1) control: distilled water; (2) NaCl: 100 mM NaCl + distilled water; (3) NA: 100 mM NaCl + 0.5 mM AsA; (4) NL: 100 mM NaCl + 0.25 mM LYC; and (5) NLA: 100 mM NaCl + 0.25 mM LYC + 0.5 mM AsA. A completely randomized block design with four replications per treatment and five plants per replication was used. The nutrient solution was oxygenated throughout the experiment, with sampling on the third day of treatment [28].

### 2.2. Determination of Growth Indicators

Aboveground and belowground relative growth rates (RGRs) were calculated following the method described by Van [29]. Root activity was determined using the triphenyltetrazolium chloride (TTC) method, as outlined by Li [30].

### 2.3. Ion Content and Transcriptional Expression Assay of Key Genes of SOS Signaling Pathway

K^+^, Ca^2+^, Na^+^, and Mg^2+^ ions were quantified using inductively coupled plasma emission spectrometry (ICP-OES, Agilent, Santa Clara, CA, USA). Cl^−^ content was determined following Nazar’s method [31]. The ion-selective transport ratios [S _Na, X_ = leaf (X/Na^+^)/root (X/Na^+^)] were calculated according to Epstein’s method [32], with X representing Ca^2+^, K^+^, and Mg^2+^, respectively. 

Gene expression levels of salt overly sensitive 1,2,3 (*SOS1*, *SOS2*, *SOS3*), Na^+^/H^+^ antiporter 1, 2, 3 (*NHX1*, *NHX2*, *NHX3*), high-affinity potassium transporter protein (*HKT1;2*), pyrophosphate-energized vacuolar membrane proton pump (*VP1*), and chloride channel (*CLC*) were determined following Livak et al. [33], with the primers used detailed in Table 1.

### 2.4. Proline (Pro) Content and Its Anabolic Key Enzyme Activities and Gene Expression Assays

Proline (Pro) content was quantified using the acid ninhydrin colorimetry method described by de Freitas [34]. The activity of Δ1-pyrroline-5-carboxylate synthase (P5CS) was assessed following the protocol of Song et al. [35], while the activity of ornithine-δ-aminotransferase (OAT) was measured according to Kim et al. [36]. Proline dehydrogenase (ProDH) activity was determined using the method outlined by Lutts [37]. A gene expression analysis of key enzymes involved in Pro synthesis and metabolism was performed using qRT-PCR, with the primer sequences provided in Table 1.

### 2.5. Ascorbic Acid (AsA) Content and Its Anabolic Key Enzyme Activities and Gene Expression Assays

Ascorbic acid (AsA) and dehydroascorbic acid (DHA) levels were measured following the protocol outlined by Jiang et al. [38]; l-galactose dehydrogenase (GalDH) activity was determined using the method described by Gatzek [39]; L-galactono-1, 4-lactone dehydrogenase (GalLDH) activity was assessed according to Ôba [40]; and ascorbate oxidase (AAO) activity was determined following the procedure by Esaka [41]. The gene expression of *GalDH*, *GalLDH*, and *AAO* was analyzed using qRT-PCR, with primer sequences detailed in Table 1.

### 2.6. Gene Expression Analysis

The total RNA from tomato leaves was extracted using the Trizol method and reverse-transcribed into cDNA using the Hyper ScriptTM III RT SuperMix (EnzyArtisan Biotech, Shanghai, China), following the manufacturer’s instructions. qPCR amplification was performed in real-time using 2 × S6 Universal SYBR qPCR Mix (enzyme Biotech, China). Each sample was run in triplicate, and each gene was analyzed with three biological and technical replicates. The relative gene expression was calculated using the 2^−ΔΔCt^ method. The qRT-PCR amplification primers are listed in Table 1. The tomato actin gene served as an internal control [42,43].

### 2.7. Statistical Analysis

Data were processed and statistically analyzed using Microsoft Excel 2020 and SPSS 19.0. Graphs were generated using Origin 2021 software. One-way analysis of variance (ANOVA) followed by Duncan’s multiple range test was utilized to ascertain the significance of differences between treatments (*p* < 0.05). The results are expressed as mean ± standard deviation.

## 3. Results

### 3.1. Exogenous AsA Promotes the Growth of Tomato Seedlings under Salt Stress

Figure 1 demonstrates that 100 mM NaCl stress significantly hindered the growth of tomato seedlings, as indicated by a notable decrease in the relative growth rate of both aerial and underground parts, along with the root activity of tomato seedlings compared to the control. However, the application of exogenous AsA mitigated the adverse effects of NaCl stress on the relative growth rate of both aerial and underground parts, as well as the root activity of tomato seedlings, resulting in significant increases of 186.1%, 141.9%, and 66.9%, respectively. In contrast, LYC application (NL treatment) exacerbated the inhibitory effect of NaCl stress on tomato seedling growth. The NLA treatment significantly reversed the aforementioned indices compared to the NL treatment. These results indicate that AsA can promote the growth of tomato seedlings under salt stress.

### 3.2. Exogenous AsA Increases the Content of Endogenous AsA of Tomato Seedlings under Salt Stress

As depicted in Figure 2, exposure to NaCl stress significantly reduced the endogenous AsA content, total AsA content, and the AsA/DHA ratio, while significantly increasing the DHA content compared to the control. The application of exogenous AsA (NA treatment) resulted in a significant increase in AsA content by 172.9%, total AsA content by 53.1%, and the AsA/DHA ratio by 338.7%, while significantly decreasing the DHA content by 37.8% compared to the NaCl treatment. Conversely, the NL treatment significantly reduced the AsA and total AsA content, as well as the AsA/DHA ratio, compared to the NaCl treatment, with a significant increase in DHA content in tomato leaves. The NLA treatment significantly decreased the DHA content and significantly increased the AsA and total AsA content, as well as the AsA/DHA ratio, compared to the NL treatment. These results indicate that exogenous AsA can increase the content of endogenous AsA and maintain a high redox pool of AsA.

### 3.3. Exogenous AsA Affects Key Enzyme Activities and the Gene Expression of AsA Anabolism of Tomato Seedlings under Salt Stress

As shown in Figure 3, the activities of GalDH and GalLDH in tomato seedling leaves under NaCl stress were significantly reduced, while AAO activity was significantly increased compared to the control, exhibiting similar trends in *AAO*, *GalDH*, and *GalLDH* gene expression. The exogenous application of AsA significantly increased the enzyme activities of GalDH and GalLDH and their gene expression, while significantly decreasing AAO activity and gene expression levels compared to the NaCl treatment. Conversely, the exogenous application of LYC significantly decreased GalDH and GalLDH activities and *GalDH* gene expression levels compared to the NaCl treatment, had no significant effect on AAO activity, but significantly upregulated *AAO* gene expression levels throughout the treatment. The application of AsA on top of the NL treatment significantly reversed these indices, increasing GalDH and GalLDH activities by 103.1% and 129.3%, respectively, and significantly decreasing AAO activity by 90.7%. These results indicate that exogenous AsA can significantly affect the activity of key enzymes and the gene expression of endogenous AsA anabolic metabolism.

### 3.4. Exogenous AsA Alleviates the Ionic Imbalance of Tomato Seedlings under Salt Stress

Compared to the control, the NaCl treatment significantly increased the Na^+^ and Cl^−^ contents as well as Na^+^/K^+^, Na^+^/Ca^2+^, and Na^+^/Mg^2+^ ratios in tomato leaves and roots, and significantly decreased K^+^, Ca^2+^, and Mg^2+^ contents, as shown in Figure 4. Treatment with exogenous AsA under salt stress significantly reduced the Na^+^ and Cl^−^ contents in both leaves and roots by 2.6% and 14.1%, and 21.6% and 16.6%, respectively, compared to the NaCl treatment. Furthermore, it significantly increased the K^+^, Ca^2+^, and Mg^2+^ contents by 10.5% and 13.6%, 36.4% and 52.6%, and 15.8% and 7.8%, respectively. Additionally, the ratios of Na^+^/K^+^, Na^+^/Ca^2+^, and Na^+^/Mg^2+^ showed a significant decrease. The NL treatment led to an accumulation of Na^+^ and Cl^−^ and a significant reduction in K^+^, Ca^2+^, and Mg^2+^ contents in both leaves and roots of tomato seedlings under NaCl treatment, further disrupting the ionic homeostasis (resulting in significantly increased Na^+^/K^+^, Na^+^/Ca^2+^, and Na^+^/Mg^2+^ ratios). However, the application of AsA significantly mitigated the negative effects of LYC on the leaves and roots of tomato seedlings under salt stress. The above indicates that exogenous AsA can reduce the ion imbalance caused by salt stress in tomato seedlings.

### 3.5. Exogenous AsA Affects the Ion Selective Absorption and Transportation Capacity of Tomato Seedlings under Salt Stress

As shown in Figure 5, compared to the control, salt stress significantly increased S_K, Na_, S_Mg, Na_, and S_Ca, Na_ by 52.6%, 28.4% and 110.5%, respectively. The exogenous spraying of AsA significantly decreased S_K, Na_, S_Mg, Na_, and S_Ca, Na_ in tomato seedlings under salt stress. Conversely, the exogenous spraying of LYC significantly decreased S_K, Na_ and S_Mg, Na_ but had no significant effect on S_Ca, Na_ under salt stress. Compared to the NL treatment, the NLA treatment significantly increased S_K, Na_ and S_Mg, Na_, while significantly decreasing S_Ca, Na_. These results indicate that exogenous AsA can significantly affect the ion selective transport ratio of tomato under salt stress.

### 3.6. Exogenous AsA Regulates the Expression of Genes Related to the SOS Pathway of Tomato Seedlings under Salt Stress

The expression levels of the *SOS1*, *SOS2*, *NHX2*, *NHX3*, *HKT1;2*, *VP1*, and *CLC* genes were downregulated to varying degrees in tomato seedling leaves under NaCl treatment compared with control, as shown in Figure 6. The exogenous spraying of AsA significantly upregulated the expression levels of the *SOS1*, *SOS2*, *NHX2*, *NHX3*, *HKT1;2*, *VP1*, and *CLC* genes in tomato seedling leaves by 2.12-, 3.35-, 2.64-, 2.19-, 3.05-, 4.55-, 5.55-, 3.76, and 1.44-fold, respectively, compared to the NaCl treatment. The expression of the above genes decreased to different degrees in the NL treatment compared to NaCl stress. However, the NLA treatment reversed this phenomenon, and all the above-mentioned genes were significantly upregulated. This indicates that exogenous AsA can regulate ion homeostasis by regulating the expression of genes related to the SOS pathway.

### 3.7. Exogenous AsA Regulates Proline (Pro) Content and Its Anabolic Key Enzyme Activities and Gene Expression in Tomato Seedlings under Salt Stress

As shown in Figure 7 and Figure 8, salt stress significantly increased Pro content and the activity and gene expression of P5CS and OAT, while significantly decreasing ProDH activity and gene expression in tomato seedling leaves compared with the control. Compared with the NaCl treatment, the NA treatment significantly reduced Pro content as well as the activities of P5CS and OAT by 60.2%, 45.6, and 14.8%, respectively. The NA treatment downregulated the expression of the P5CS gene by 0.25-fold and the expression of the *OAT* gene by 0.38-fold under NaCl stress. Additionally, the NA treatment significantly increased ProDH activity by 20.7% and upregulated *ProDH* gene expression by 2.64-fold. However, the exogenous application of the AsA inhibitor LYC significantly increased proline content as well as OAT and P5CS activities and their gene expression in tomato seedling leaves under salt stress conditions, while significantly decreasing ProDH activity and gene expression. The exogenous spraying of AsA on top of the NL treatment then significantly reversed the trend of the above indicators. These results indicate that exogenous AsA can reduce the proline content in tomato under salt stress by regulating the activity of key enzymes and gene expression in proline anabolism.

### 3.8. Analysis of Correlation

From the results of the correlation analysis, significant positive or negative correlations were observed between some salt-responsive physiological and morphological parameters (Figure 9). For example, aboveground and belowground relative growth rates were significantly and negatively correlated with Cl^−^ content in plant leaves and roots, and the salt over-sensitive (SOS) regulatory pathways (*SOS1*, *SOS2*, and *SOS3* gene expression) were significantly and positively correlated with the expression of the *ProDH* and *GalDH* genes. This suggests that some of the parameters may have similar responses to salt stress. However, the significant correlation between two parameters does not imply that they can be substituted for each other.

### 3.9. Mechanism of AsA in Alleviating Salt Stress

As shown in Figure 10, salt stress induces a high oxidative state in cells, disrupts ionic homeostasis in the roots and leaves of tomato seedlings, accumulates large amounts of Na^+^ and Cl^−^, and reduces the expression of genes related to the SOS pathway, thereby disrupting the dynamic balance of proline. These effects ultimately lead to decreased root activity and affect the overall growth and development of the plant. However, the exogenous spraying of AsA significantly reverses these phenomena. Exogenous AsA effectively improves the growth characteristics of plants under salt stress by maintaining high endogenous AsA levels and redox pools. This maintenance mediates the SOS pathway to alleviate ionic toxicity and mitigates osmotic stress by regulating proline synthesis and metabolism. These actions enhance the plant’s resilience to stress.

## 4. Discussion

Salt stress, as a major abiotic stressor, induces ion toxicity, osmotic stress, and oxidative stress, leading to physiological imbalances in plants. These effects severely restrict crop growth, development, and yield, posing a global threat to agricultural production [44]. Plant roots are particularly sensitive to stress signals, and salt stress inhibits their growth, reducing root vitality and impacting the absorption of water and nutrients, thereby affecting the entire plant’s normal growth. The findings of this study demonstrate that salt stress significantly inhibited the growth and biomass accumulation of tomato seedlings. However, the exogenous application of AsA effectively improved plant growth characteristics under salt stress by enhancing root vitality, as shown in Figure 1. These results are consistent with earlier reports indicating that AsA can mitigate the adverse effects of salt stress on plant biomass. Furthermore, we observed that exogenous AsA application to tomato seedlings under salt stress increased endogenous AsA levels and the AsA/DHA ratio, as illustrated in Figure 2.

Research has shown that the cellular redox state is an important factor in plants’ resistance to abiotic stress [45]. Maintaining a high redox pool of AsA (AsA/DHA ratio) is crucial for plants to scavenge excessive ROS and keep the thiol groups of soluble proteins and membrane proteins in a reduced state. The results indicate that the exogenous application of AsA can effectively promote the growth of tomato seedlings under salt stress, and is positively correlated with endogenous AsA levels (Figure 9). The L-galactose pathway is the main pathway for plants to synthesize AsA, and GalLDH and GalDH are key enzymes in the final two steps of AsA synthesis in the L-galactose pathway. Studies in melon [46] and tobacco [47] have found a close correlation between the content of endogenous AsA and the activity and gene expression of GalLDH; a significant correlation between GalDH activity and AsA content was found in shepherd’s purse and Arabidopsis [48]. Studies on corn extracts have found that lycorine (LYC) is an effective inhibitor of GalLDH [49]. To further explore the role of AsA, we applied LYC under salt stress and found that the activities and gene expression of GalLDH and GalDH in tomato seedlings were inhibited, as were the endogenous AsA levels and AsA/DHA ratio (Figure 1 and Figure 3). However, after the exogenous application of AsA, the activities and gene expression of GalDH and GalLDH were significantly increased (Figure 3). AAO is a key enzyme in the AsA oxidation metabolism pathway. When studying the salt tolerance of AAO transgenic plants, it was found that under normal conditions, there was no obvious phenotype change in AAO overexpressing and antisense transgenic plants compared to the control. However, under high salt conditions, the germination and photosynthetic rates of the antisense AAO transgenic plants were higher compared to those of the wild type and AAO overexpressing transgenic plants. Furthermore, it was found that the H_2_O_2_ content in overexpressing transgenic plants was the highest, while the AsA content was very low, indicating that inhibiting the expression of the AAO gene can increase the salt tolerance of plants [50]. This study found that salt stress led to a significant increase in AAO activity and gene expression in tomato leaves, while the exogenous spraying of AsA significantly reduced the AAO activity and downregulated AAO expression in tomato leaves under NaCl treatment and NL treatment (Figure 3). These findings suggest that the exogenous AsA maintains a high level of endogenous AsA and redox pool by altering the activities and gene expression levels of enzymes involved in AsA synthesis and metabolism, thus protecting plants from oxidative damage caused by salt stress (NaCl and NL treatments) and improving the salt tolerance of tomato seedlings.

Plants maintain ion homeostasis in response to stress environments by reducing excess Na^+^ (reducing Na^+^ uptake, promoting Na^+^ efflux, and compartmentalizing Na^+^) to enhance salt tolerance. Na^+^ efflux and compartmentalization are active transport processes mainly driven by the proton gradient generated by H^+^-ATPase and H^+^-PPase, and mediated by Na^+^/H^+^ antiporters [51]. Besides Na^+^, Cl^−^ efflux and compartmentalization are also important salt tolerance mechanisms. Studies have shown that exogenous silicon can reduce Na^+^ content in tomato seedling leaves, promote the uptake of K^+^, Ca^2+^, and Mg^2+^, and increase the K^+^/Na^+^ and Ca^2+^/Na^+^ ratios in leaves by at least 2-fold [52]; exogenous melatonin significantly increases the expression of *SOS* genes (*SOS1*, *SOS2*, and *SOS3*) in plants under salt stress, reducing Na^+^ content in the aboveground parts and increasing the K^+^/Na^+^ ratio [53,54]; exogenous boron, by upregulating the expression of *CLC* genes, reduces Cl^−^ uptake, alleviating the impact of NaCl stress on beet growth [55]. This study found that salt stress disrupted ion homeostasis in tomato seedling roots and leaves, resulting in the accumulation of Na^+^ and Cl^−^, while reducing the uptake of K^+^, Ca^2+^, and Mg^2+^, affecting the absorption and transport ratios of ions in roots and leaves (Figure 4 and Figure 5), and decreasing the expression of the *SOS* (*SOS1*, *SOS2*, and *SOS3*), *HKT1;2*, *NHX* (*NHX1*, *NHX2*, and *NHX3*), *CLC*, and *VP1* genes in seedling leaves (Figure 6). Similar conclusions have been drawn in cabbage [56], cucumber [57], eggplant [58], and other plants. However, spraying AsA under salt stress and NL treatment revealed that exogenous AsA enhanced Na^+^ and Cl^−^ efflux by upregulating the expression of SOS and CLC genes. This treatment also promoted the accumulation of K^+^, Ca^2+^, and Mg^2+^, resulting in reduced Na^+^/K^+^, Na^+^/Ca^2+^, and Na^+^/Mg^2+^ ratios. Additionally, it reduced Na^+^ selectivity in aboveground parts while enhancing the selective absorption and transport capacity of K^+^, Ca^2+^, and Mg^2+^ from roots to leaves (Figure 4 and Figure 5). Regulating *NHX* gene expression compartmentalized Na^+^ into vacuoles, a key mechanism for maintaining water absorption in plants under salt stress. Regulating *HKT* gene expression recycles Na^+^ from transpiration flow to avoid the excessive accumulation of Na^+^ in photosynthetic tissues. Upregulating *VP1* gene expression provides more driving force for ion transmembrane transport to ensure the normal operation of Na^+^/H^+^ antiporters, thereby enhancing plant salt tolerance (Figure 6).

Proline, as an osmoprotectant, membrane stabilizer, and ROS scavenger [59,60,61,62], and exogenous AsA may have different regulatory patterns in plants under different abiotic stresses, with Pro content increasing [63,64] or decreasing [65,66]. Kavi [67] proposed that maintaining a dynamic balance of Pro under stress conditions is a necessary condition for normal plant growth and development. Therefore, evaluating the resistance of plants to abiotic stress by Pro content is of great significance. In this study, NaCl stress increased the activity and gene expression of key enzymes in Pro synthesis pathways (P5CS and OAT) in tomato seedling leaves while inhibiting the activity and transcription level of the rate-limiting enzyme in the Pro degradation pathway (ProDH), resulting in a significant increase in Pro content in leaves, indicating that plants have initiated defense mechanisms to resist osmotic stress caused by salt stress (Figure 7 and Figure 8). This is consistent with studies in papaya [68], cucumber [69], and tomato [70]. However, some studies have shown that excess Pro not only fails to alleviate stress-induced damage but also exacerbates growth inhibition under stress [71]. In this study, it was found that the exogenous spraying of the inhibitor LYC dramatically increased Pro content in tomato seedling leaves under salt stress, but the exogenous spraying of AsA reduced excess Pro content in leaves by reducing the synthesis pathway of Pro and enhancing the degradation pathway of Pro, maintaining the dynamic balance of Pro in plants, thereby enhancing the salt tolerance of tomato seedlings.

## 5. Conclusions

In conclusion, salt stress induces ion imbalance and osmotic stress in tomato seedlings, severely affecting their growth. However, the application of exogenous AsA significantly improves the growth characteristics of plants under salt stress conditions. The research results indicate that exogenous AsA, by regulating the activity and gene expression levels of enzymes related to endogenous AsA synthesis, increases the intracellular AsA content, maintaining the intracellular redox state in a reduced state. Furthermore, exogenous AsA also regulates the transcription levels of multiple ion transporters through the SOS pathway, enhancing the plant’s selective absorption of K^+^, Ca^2+^, and Mg^2+^, thereby alleviating ion imbalance caused by salt stress. Meanwhile, exogenous AsA alleviates osmotic stress by regulating Pro synthesis and metabolism, enhancing the salt tolerance of tomato seedlings (Figure 10).

## Figures and Tables

**Figure 1 plants-13-01672-f001:**
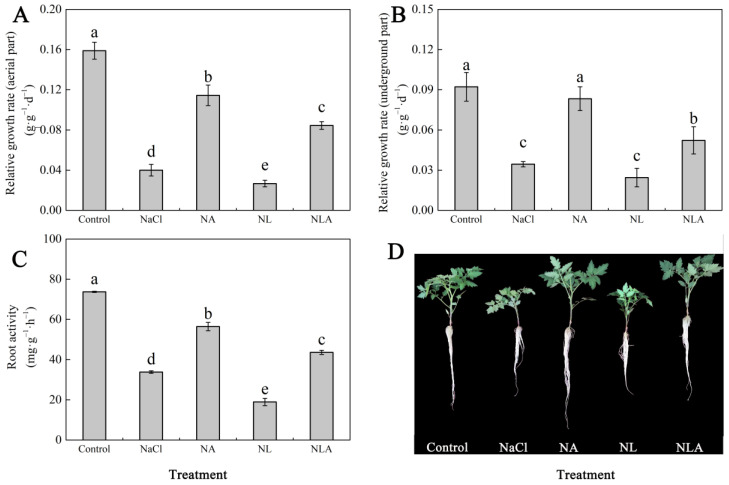
Values of the relative growth rate (aerial part) (**A**), relative growth rate (underground part) (**B**), root activity (**C**), and photographs of tomato seedlings on the third day after different treatments (**D**) in the leaves of salt-stressed tomato seedlings with or without exogenous reduced ascorbic acid (AsA) and Lycorine (LYC, AsA synthesis inhibitor) spraying. All measurements were performed on leaves at 3 d after treatment. Error bars represent SD (*n* = 4). Different letters indicate significance differences among treatments (*p* < 0.05). Control: no NaCl and no AsA and no LYC; NaCl: 100 mM NaCl; NA: 100 mM NaCl + 0.5 mM AsA; NL: 100 mM NaCl + 0.25 mM LYC; NLA: 100 mM NaCl + 0.25 mM LYC + 0.5 mM AsA.

**Figure 2 plants-13-01672-f002:**
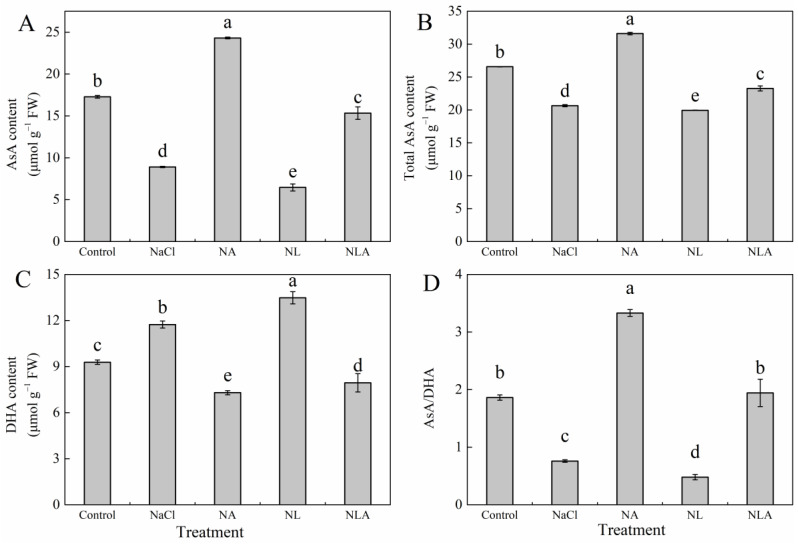
Values of the AsA content (**A**), total AsA content (**B**), DHA content (**C**), and ratio of AsA/DHA (**D**) in the leaves of salt-stressed tomato seedlings with or without exogenous reduced ascorbic acid (AsA) and Lycorine (LYC, AsA synthesis inhibitor) spraying. All measurements were performed on leaves at 3 d after treatment. Error bars represent SD (*n* = 4). Different letters indicate significance differences among treatments (*p* < 0.05). Control: no NaCl and no AsA and no LYC; NaCl: 100 mM NaCl; NA: 100 mM NaCl + 0.5 mM AsA; NL: 100 mM NaCl + 0.25 mM LYC; NLA: 100 mM NaCl + 0.25 mM LYC + 0.5 mM AsA.

**Figure 3 plants-13-01672-f003:**
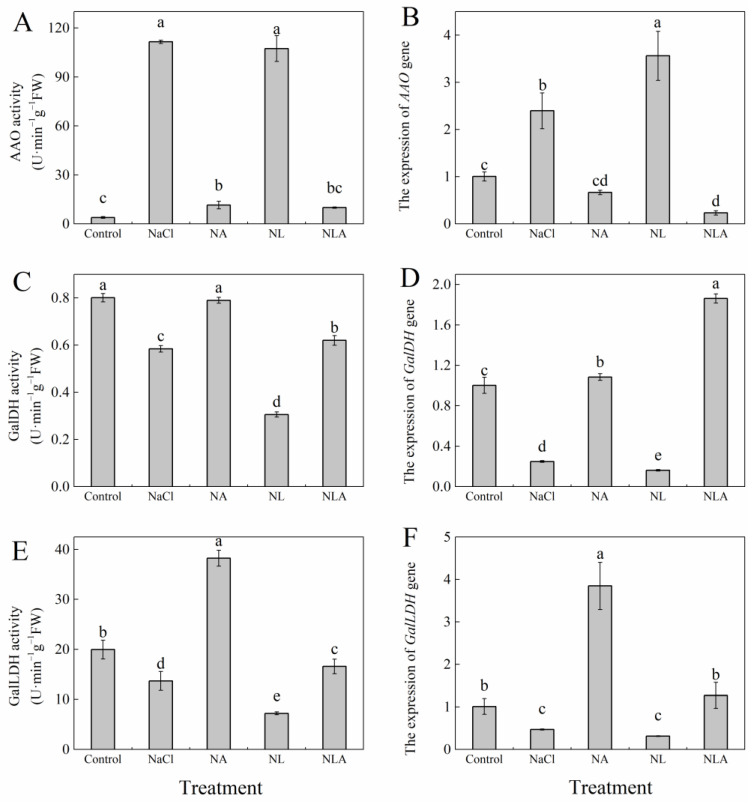
Values of the AAO activity (**A**), *AAO* gene expression (**B**), GalDH activity (**C**), *GalDH* gene expression (**D**), GalLDH activity (**E**), and *GalLDH* gene expression (**F**) in the leaves of salt-stressed tomato seedlings with or without exogenous reduced ascorbic acid (AsA) and Lycorine (LYC, AsA synthesis inhibitor) spraying. All measurements were performed on leaves at 3 d after treatment. Error bars represent SD (*n* = 4). Different letters indicate significance differences among treatments (*p* < 0.05). Control: no NaCl and no AsA and no LYC; NaCl: 100 mM NaCl; NA: 100 mM NaCl + 0.5 mM AsA; NL: 100 mM NaCl + 0.25 mM LYC; NLA: 100 mM NaCl + 0.25 mM LYC + 0.5 mM AsA.

**Figure 4 plants-13-01672-f004:**
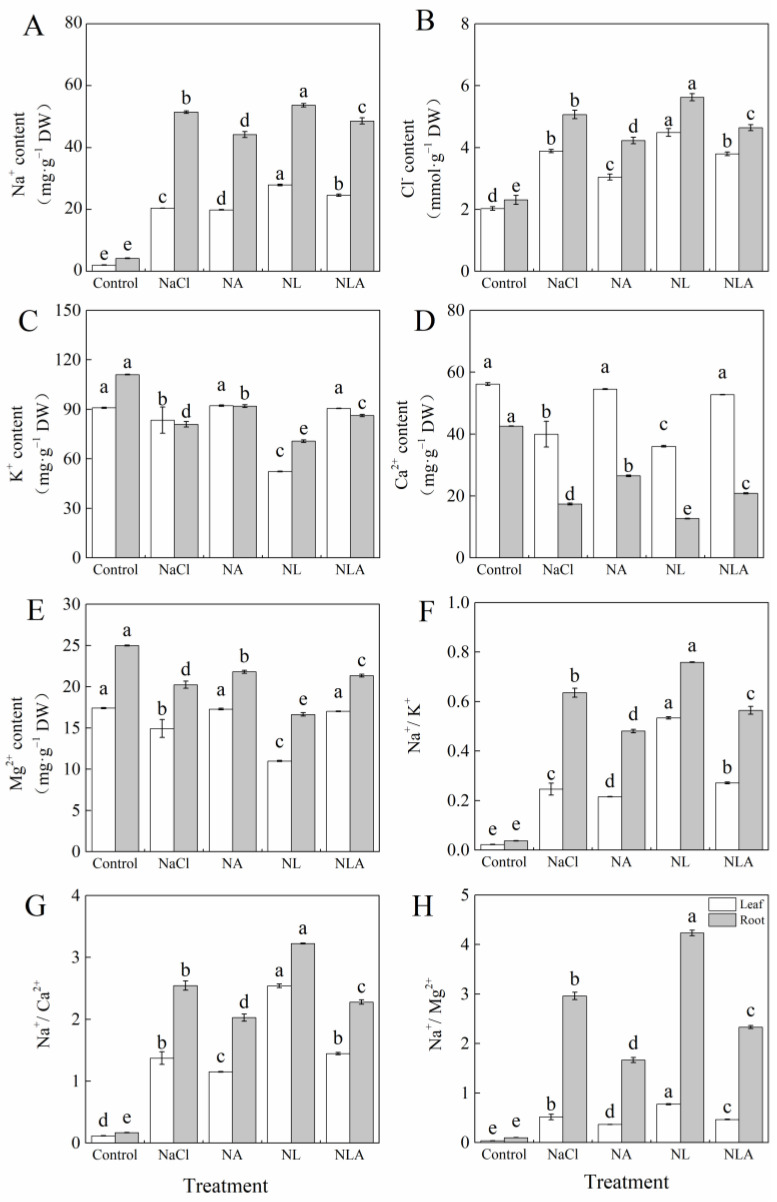
Values of the Na^+^ content (**A**), Cl^−^ content (**B**), K^+^ content (**C**), Ca^2+^ content (**D**), Mg^2+^ content (**E**), Na^+^/K^+^ (**F**), Na^+^/Ca^2+^ (**G**) and Na^+^/Mg^2+^ (**H**) in the leaves and roots of salt-stressed tomato seedlings with or without exogenous reduced ascorbic acid (AsA) and Lycorine (LYC, AsA synthesis inhibitor) spraying. All measurements were performed on leaves at 3 d after treatment. Error bars represent SD (*n* = 4). Different letters indicate significance differences among treatments (*p* < 0.05). Control: no NaCl and no AsA and no LYC; NaCl: 100 mM NaCl; NA: 100 mM NaCl + 0.5 mM AsA; NL: 100 mM NaCl + 0.25 mM LYC; NLA: 100 mM NaCl + 0.25 mM LYC + 0.5 mM AsA.

**Figure 5 plants-13-01672-f005:**
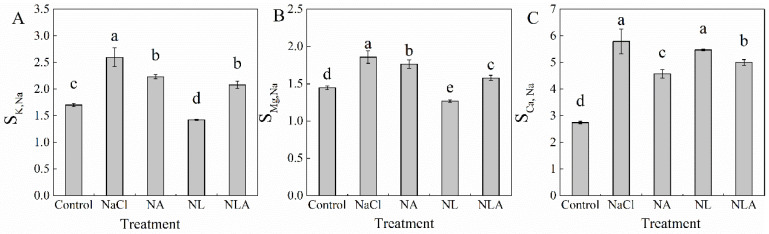
Values of S_K, Na_ (**A**), S_Mg, Na_ (**B**), and S_Ca, Na_ (**C**) in leaves of salt-stressed tomato seedlings with or without exogenous reduced ascorbic acid (AsA) and Lycorine (LYC, AsA synthesis inhibitor) spraying. All measurements were performed on leaves at 3 d after treatment. Error bars represent SD (*n* = 4). Different letters indicate significance differences among treatments (*p* < 0.05). Control: no NaCl and no AsA and no LYC; NaCl: 100 mM NaCl; NA: 100 mM NaCl + 0.5 mM AsA; NL: 100 mM NaCl + 0.25 mM LYC; NLA: 100 mM NaCl + 0.25 mM LYC + 0.5 mM AsA.

**Figure 6 plants-13-01672-f006:**
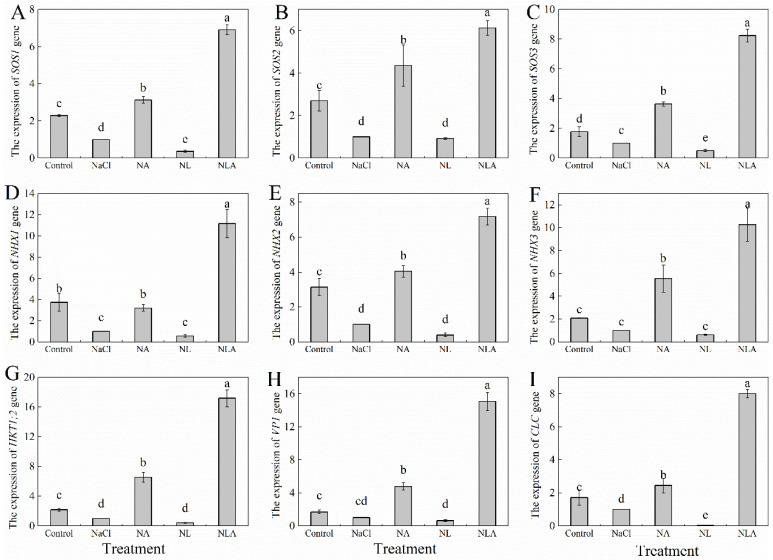
Expression of *SOS1* (Salt overly sensitive 1 gene) (**A**), *SOS2* (Salt overly sensitive 2 gene) (**B**), *SOS3* (Salt overly sensitive 3 gene) (**C**), *NHX1* (Na^+^/H^+^ antiporter 1 gene) (**D**), *NHX2* (Na^+^/H^+^ antiporter 2 gene) (**E**), *NHX3* (Na^+^/H^+^ antiporter 3 gene) (**F**), *HKT1;2* (high-affinity potassium transporter protein gene) (**G**), *VP1* (pyrophosphate-energized vacuolar membrane proton pump gene) (**H**), and *CLC* (chloride channel protein gene) (**I**) genes in the leaves of salt-stressed tomato seedlings with or without exogenous reduced ascorbic acid (AsA) and Lycorine (LYC, AsA synthesis inhibitor) spraying. All measurements were performed on leaves at 3 d after treatment. Error bars represent SD (*n* = 3). Different letters indicate significance differences among treatments (*p* < 0.05). Control: no NaCl and no AsA and no LYC; NaCl: 100 mM NaCl; NA: 100 mM NaCl + 0.5 mM AsA; NL: 100 mM NaCl + 0.25 mM LYC; NLA: 100 mM NaCl + 0.25 mM LYC + 0.5 mM AsA.

**Figure 7 plants-13-01672-f007:**
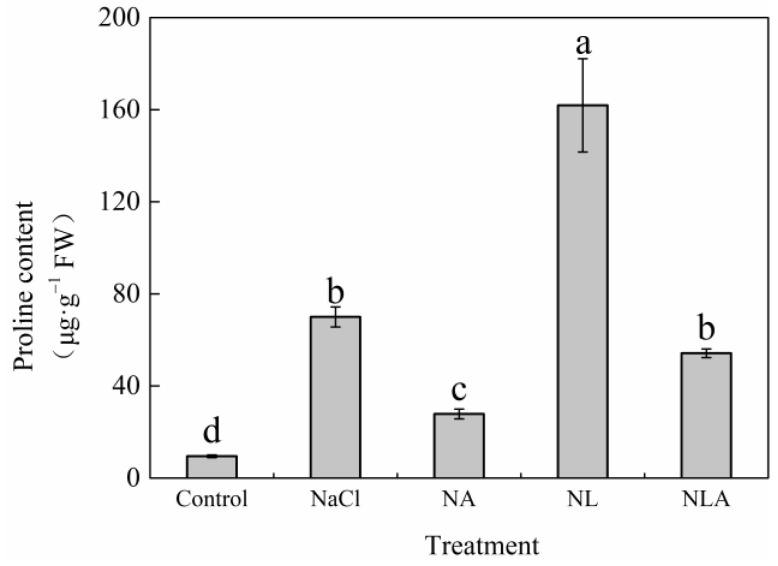
Values of proline content in the leaves of salt-stressed tomato seedlings with or without exogenous reduced ascorbic acid (AsA) and Lycorine (LYC, AsA synthesis inhibitor) spraying. All measurements were performed on leaves at 3 d after treatment. Error bars represent SD (*n* = 4). Different letters indicate significance differences among treatments (*p* < 0.05). Control: no NaCl and no AsA and no LYC; NaCl: 100 mM NaCl; NA: 100 mM NaCl + 0.5 mM AsA; NL: 100 mM NaCl + 0.25 mM LYC; NLA: 100 mM NaCl + 0.25 mM LYC + 0.5 mM AsA.

**Figure 8 plants-13-01672-f008:**
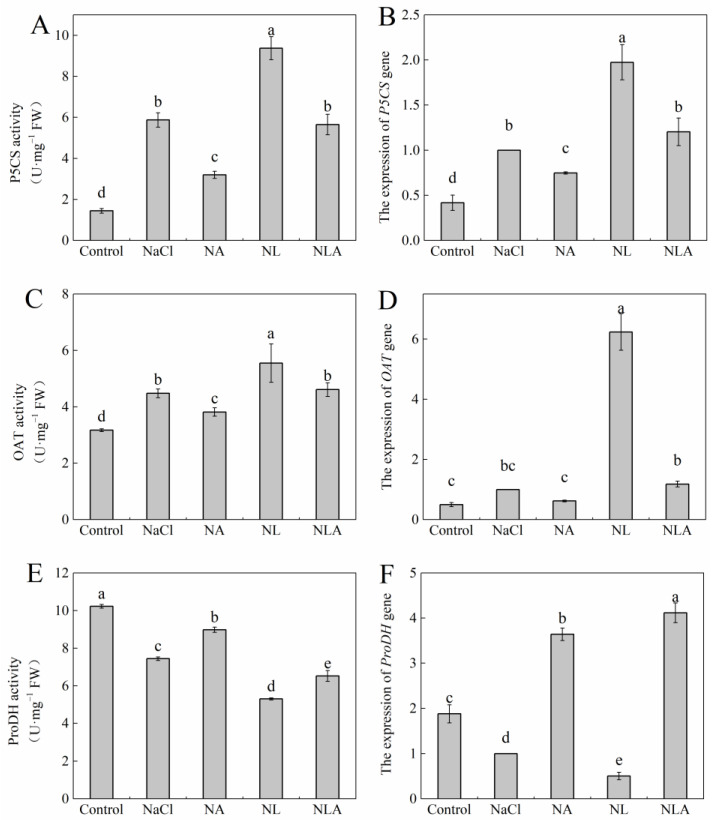
Values of Δ1-pyrroline-5-carboxylate synthase (P5CS) activity (**A**), expression of *P5CS* (Δ1-pyrroline-5-carboxylate synthase gene) gene (**B**), ornithine-δ-aminotransferase (OAT) activity (**C**), expression of *OAT* (ornithine-δ-aminotransferase gene) gene (**D**), proline dehydrogenase (ProDH) activity (**E**), and expression of *ProDH* (proline dehydrogenase gene) gene (**F**) in the leaves of salt-stressed tomato seedlings with or without exogenous reduced ascorbic acid (AsA) and Lycorine (LYC, AsA synthesis inhibitor) spraying. All measurements were performed on leaves at 3 d after treatment. Error bars represent SD (*n* = 4). Different letters indicate significance differences among treatments (*p* < 0.05). Control: no NaCl and no AsA and no LYC; NaCl: 100 mM NaCl; NA: 100 mM NaCl + 0.5 mM AsA; NL: 100 mM NaCl + 0.25 mM LYC; NLA: 100 mM NaCl + 0.25 mM LYC + 0.5 mM AsA.

**Figure 9 plants-13-01672-f009:**
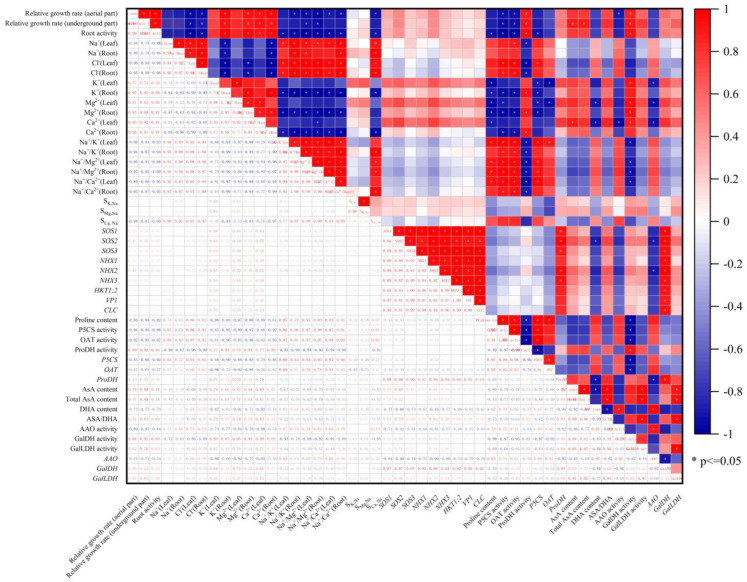
The linear regression coefficients among 48 salt response parameters.

**Figure 10 plants-13-01672-f010:**
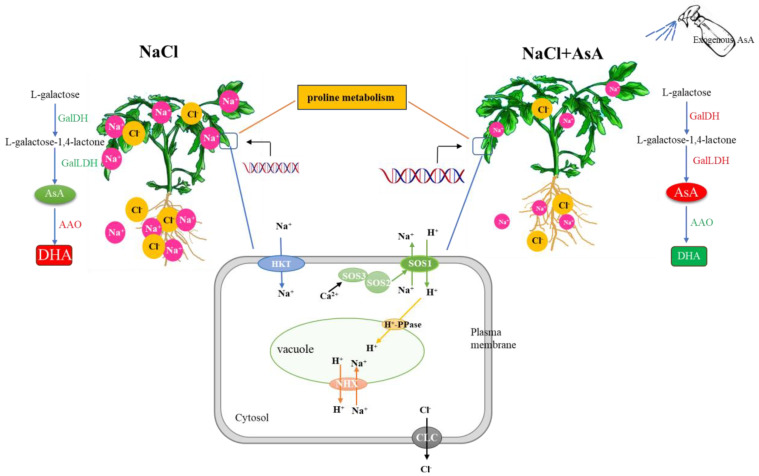
Schematic representation of the mechanism by which exogenous AsA alleviates salt stress in tomato seedlings.

**Table 1 plants-13-01672-t001:** Quantitative real-time PCR sequences.

Gene	Primer	Sequence (5′ to 3′)
*Actin* (NM_001323002.1)	FORWARD	TGGTCGGAATGGGAAAG
	REVERSE	CTCAGTCAGGAGAACAGGGT
*SOS1* (AJ717346.1)	FORWARD	GCTGATGTCTCTGGTGTCTTGACTG
	REVERSE	TTGATGACTCTCGCCCTTGAAAGC
*SOS2* (NM_001247281.2)	FORWARD	TATTTCCCGCCAACCTGCTAAAGTC
	REVERSE	GACCAGCCCTATTTGCCGTTACC
*SOS3* (AJ717347.1)	FORWARD	TATTCCACCCAAATGCACCAGTAGC
	REVERSE	CATTCAGCAGCGCCAAAACCATC
*NHX1* (NM_001246987.1)	FORWARD	CTTGGTCTGGTTCTGGTTGGAAGG
	REVERSE	AGCCCACCATATCGTGACCTGTAG
*NHX2* (NM_001328634.1)	FORWARD	TCACTGCTACCACTGCCATTGTTG
	REVERSE	ACCATCACCCACAACTTCCAAAGC
*NHX3* (NM_001247326.2)	FORWARD	TGGTTGGAAGGGCAGCATTTGTC
	REVERSE	TGAAACAGCACCTCGCATAAGTCC
*HKT1;2* (NM_001302904.1)	FORWARD	CCTACCGTCTTTTCGTCCTCA
	REVERSE	GCTTCCCCACCAAGAAACATC
*VP1* (NM_001278976.2)	FORWARD	GATGGTTGAGGAAGTGCGTAGGC
	REVERSE	CACAGGTGGCATAGTCAGGCTTG
*CLC* (NM_001247096.2)	FORWARD	CGTCGCCTTCGCCTTCTAATCG
	REVERSE	CAACAAGCAACATCGCCCATTCC
*P5CS* (NM_001246978.2)	FORWARD	TGGAAGATTAGGAGCCCTCTGTGAG
	REVERSE	CTAAGCCGCTGACGACCAACAC
*OAT* (NM_001247674.3)	FORWARD	GGCTCTCATTGTCTCGTGCTGTG
	REVERSE	GGGCAACCGAATCTCCAAAATCAAC
*ProDH* (NM_001347105.1)	FORWARD	CCACCACCACGACCATCACAAC
	REVERSE	CATTACCCACATGCCCAAATCAACC
*AAO* (NM_001247900.2)	FORWARD	ACAAGCAGGACTACAAGGAATGATG
	REVERSE	AGGCAATGAAGCAAGACCAGTTG
*GalDH* (XM_004230609.4)	FORWARD	CAACGACTGGAATGGACGAAGAAG
	REVERSE	AACAGGAGATCACAATTCACAAGACC
*GalLDH* (NM_001247674.3)	FORWARD	GTTGAGAGGCAGGAGCTTGTAGAAC
	REVERSE	TGTCACAACCACAACGGCATCAG

Note: *Actin*: Actin gene; *SOS1*: Salt overly sensitive 1 gene; *SOS2*: Salt overly sensitive 2 gene; *SOS3*: Salt overly sensitive 3 gene; *NHX1*: Na^+^/H^+^ antiporter 1 gene; *NHX2*: Na^+^/H^+^ antiporter 2 gene; *NHX3*: Na^+^/H^+^ antiporter 3 gene; *HKT1;2*: high-affinity potassium transporter protein gene; *VP1*: pyrophosphate-energized vacuolar membrane proton pump gene; *CLC*: chloride channel protein gene; *P5CS*: Δ1-pyrroline-5-carboxylate synthase gene; *OAT*: ornithine-δ-aminotransferase gene; *ProDH*: proline dehydrogenase gene; *AAO*: ascorbate oxidase gene; *GalDH*: L-galactose dehydrogenase gene; *GalLDH*: L-galactono-1, 4-lactone dehydrogenase gene.

## Data Availability

Data is contained within the article.

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
