# Peer review of "Ascorbic Acid Improves Tomato Salt Tolerance by Regulating Ion Homeostasis and Proline Synthesis"

_plants, 2024, doi:10.3390/plants13121672_

Round 1
Reviewer 1 Report (Previous Reviewer 2)
Comments and Suggestions for Authors
Manuscript „Ascorbic acid improves tomato salt tolerance by regulating ion homeostasis and proline synthesis deccribed by Xianjun Chen, Hongwei Han, Yundan Cong, Xuezhen Li, Wenbo Zhang, Jinxia Cui, Wei Xu, Shengqun Pang, Huiying Liu ” requires minor improvement in accordance with the points below.
-
Line 167, 168, 182,185, 270-272 – the bibliography data should be completed in property way, according to Instruction for Authors
-
Lines 330, 391, 639 - no spaces
-
Line 598 - please check if the reference is correct
Author Response
Dear Reviewer 1,
We sincerely appreciate your review and valuable feedback on our manuscript. Your insightful comments have been instrumental in improving the quality of our work. Below are our responses to your specific suggestions:
- Line 167, 168, 182,185, 270-272 – the bibliography data should be completed in property way, according to Instruction for Authors
Response: Thank you very much for your meticulous review and valuable suggestions. We have carefully revised the bibliography data in lines 167, 168, 182, 185, and 270-272 to ensure they are formatted according to the Instructions for Authors.
- Lines 330, 391, 639 - no spaces
Response: Thank you for your careful review and valuable feedback. As suggested, we have revised the manuscript to correct the spacing issues in lines 330, 391, and 639. We appreciate your attention to these details.
- Line 598 - please check if the reference is correct
Response: Thank you for your valuable feedback. We have carefully reviewed and verified the reference at line 598. Necessary corrections have been made to ensure accuracy.We appreciate your attention to this detail.
Once again, we are grateful for your constructive comments and efforts in improving our manuscript. We hope that the revised version meets your expectations.
Sincerely,
Huiying Liu PhD

Reviewer 2 Report (Previous Reviewer 1)
Comments and Suggestions for Authors
The author has addressed most of my concerns in the revised version of the manuscript. I agree that the revised version can be accepted for publication in "Plants".
Author Response
Dear Reviewer 2,
We sincerely appreciate your thorough review and valuable feedback, which have greatly contributed to the improvement of our work. Your recognition and acceptance of our article mean a lot to us.
Thank you once again for your time, effort, and support.
Sincerely,
Huiying Liu PhD
This manuscript is a resubmission of an earlier submission. The following is a list of the peer review reports and author responses from that submission.
Round 1
Reviewer 1 Report
Comments and Suggestions for Authors
1. The results section reads like a description of the experimental treatments without clearly stated conclusions. The authors need to articulate what conclusions can be drawn from each experiment.
The titles of the results sections are overly simplistic. Each title should summarize the conclusions drawn from the corresponding experiments.
2. Details in Figures: In Figure 1, how many plants were used to generate the statistical data for the experimental treatments? Specifically, for Figure 1D, how many plants contributed to the statistics presented? The letters a, b, c, d, e in Figures 1-8 indicate statistical differences, but this is not explained in the figure legends. Additionally, the figure legends are too brief and need detailed explanations. The authors should carefully revise and supplement the legends.
3. In Figure 7, the units for proline content are not specified. The authors need to indicate the units used. And need give detailed protocol of the proline content test and the Figure legend.
4. It is now common to use 2-3 reference genes for normalization in qPCR experiments. The authors should include the primer sequences listed in Table 1 as supplementary material. The qPCR experiments should clearly state the number of biological and technical replicates. Detailed information about the comparisons and analyses of qPCR results should be included in the figure legends.
Comments on the Quality of English LanguageSome editing is required throughout the manuscript to address numerous mistakes.
Reviewer 2 Report
Comments and Suggestions for Authors
Manuscript „Ascorbic acid improves tomato salt tolerance by regulating ion homeostasis and proline synthesis deccribed by Xianjun Chen, Hongwei Han, Yundan Cong, Xuezhen Li, Wenbo Zhang, Jinxia Cui, Wei Xu, Shengqun Pang, Huiying Liu ” describes interesting studies concerning regulation salt tolerances after ascorbic acid application. However some issues require clarification or improvement in accordance with the points below .
Materials and methods
-
Line 123 – the bibliography data, concerning previous experiments should be completed
-
Paragraph 1.2 and 1.3 - the methods should be described shortly. The names of genes should be written italic
-
Paragraph 1.4 - the methods of proline content quantification is very old. It should be applied newer. Why did the authors use such an old method of determination when much newer ones are available?
-
Line 165 - information about producer of system to obtain cDNA should be placed after its name
Results
-
1. Figure 4 – the explanation of bars should be placed on each diagram or in the figure description
-
Figure 10 – the part of description of Figure 10 should be placed in paragraph “Results”
-
Why do the authors not refer their results also to controls when they describe most of their experiments
-
Figure 9 should be corrected because it is unreadable.
Discussion
-
Line 322/323 bibligraphy data should be added
-
why did authors choose SOS1, SOS2, SOS3, NHX1, NHX2, NHX3, HKT1 HKT2, VP1 and CLC genes for their studies? Please explain it.